# The Interaction between Stress and Inflammatory Bowel Disease in Pediatric and Adult Patients

**DOI:** 10.3390/jcm13051361

**Published:** 2024-02-27

**Authors:** Oana Belei, Diana-Georgiana Basaca, Laura Olariu, Manuela Pantea, Daiana Bozgan, Anda Nanu, Iuliana Sîrbu, Otilia Mărginean, Ileana Enătescu

**Affiliations:** 1First Pediatric Clinic, Disturbances of Growth and Development on Children Research Center, “Victor Babeș” University of Medicine and Pharmacy, 300041 Timișoara, Romania; belei.oana@umft.ro (O.B.); marginean.otilia@umft.ro (O.M.); 2Department of Pediatrics, First Pediatric Clinic, “Victor Babeș” University of Medicine and Pharmacy, 300041 Timișoara, Romania; tunealaura@yahoo.com; 3Twelfth Department, Neonatology Clinic, “Victor Babeș” University of Medicine and Pharmacy, 300041 Timișoara, Romania; manu.pantea@gmail.com (M.P.); lena_urda@yahoo.com (I.E.); 4Clinic of Neonatology, “Pius Brânzeu” County Emergency Clinical Hospital, 300723 Timișoara, Romania; daiana.bozgan@yahoo.com; 5Third Pediatric Clinic, “Louis Țurcanu” Emergency Children Hospital, 300011 Timișoara, Romania; anda.nanu97@yahoo.com (A.N.); sirbu.iuliana18@yahoo.com (I.S.)

**Keywords:** inflammatory bowel disease, stress, anxiety, intestinal microbiota, gut–brain axis

## Abstract

**Background:** Inflammatory bowel diseases (IBDs) have seen an exponential increase in incidence, particularly among pediatric patients. Psychological stress is a significant risk factor influencing the disease course. This review assesses the interaction between stress and disease progression, focusing on articles that quantified inflammatory markers in IBD patients exposed to varying degrees of psychological stress. **Methods:** A systematic narrative literature review was conducted, focusing on the interaction between IBD and stress among adult and pediatric patients, as well as animal subjects. The research involved searching PubMed, Scopus, Medline, and Cochrane Library databases from 2000 to December 2023. **Results:** The interplay between the intestinal immunity response, the nervous system, and psychological disorders, known as the gut–brain axis, plays a major role in IBD pathophysiology. Various types of stressors alter gut mucosal integrity through different pathways, increasing gut mucosa permeability and promoting bacterial translocation. A denser microbial load in the gut wall emphasizes cytokine production, worsening the disease course. The risk of developing depression and anxiety is higher in IBD patients compared with the general population, and stress is a significant trigger for inducing acute flares of the disease. **Conclusions:** Further large studies should be conducted to assess the relationship between stressors, psychological disorders, and their impact on the course of IBD. Clinicians involved in the medical care of IBD patients should aim to implement stress reduction practices in addition to pharmacological therapies.

## 1. Introduction

### 1.1. Overview of the Epidemiology and Pathophysiology of Inflammatory Bowel Disease

The economic and financial cost of inflammatory bowel disease (IBD) is recognized as a severe, global public health problem. IBD is a broad category of complex, protracted bowel inflammation characterized by a variety of variables, including emotional distress, autonomic dysfunction, dysbiosis of the gut microbiota, and immunological modulations related to disease activity. The characterization of local and systemic immune responses in IBD and the pathways via which inflammation enters the central nervous system (CNS) as well as their effects on brain-resident immune and glial cells have become more and more the focus of recent research [1].

Significant attention should be given to the gradual increase in the prevalence of IBD in both children and adults. Currently, there remains a lack of a complete understanding regarding the underlying mechanisms of IBD. It is believed that the development of IBD involves complex interplays between genetics, environmental factors, and gut microbiota. However, the fluctuating nature of IBD, characterized by periods of relapse and remission, emphasizes the significance of additional factors, including psychological stress [2].

In this review, the authors will present up-to-date evidence discussing the impact of stress on IBD across different stages, encompassing both children and adults. IBD, encompassing ulcerative colitis (UC) and Crohn’s disease (CD), is a chronic inflammatory disorder of the intestines that affects a substantial number of individuals globally, with a relapsing and remitting course [2]. It should be noted that in the twenty-first century, IBD is quickly expanding in prevalence in rising industrial nations and is progressively becoming a global illness [3]. Even though IBD can develop at any age, it is diagnosed in 25% of individuals before the age of 20 [4]. IBD in children is more common in some nations than others, although the global trend is rising. The incidence ranges from 0.1 to 13.9/100,000 for CD, from 0.3 to 15/100,000 for UC, and between 0.5 and 23/100,000 for IBD [4,5]. Children may present with particular signs, such as poor growth and delayed puberty, in addition to the typical gastrointestinal (GI) symptoms (abdominal discomfort, diarrhea, hematochezia, and weight loss) identical to those of adults [6]. Periods of heightened symptomatology known as active phases alternate with remission phases throughout the disease, which is typically unpredictable. CD is characterized by non-continuous inflammation that extends through the entire wall of the intestine, primarily affecting the terminal ileum, caecum, colon, and perianal area. In contrast, UC is distinguished by continuous inflammation and ulceration limited to the colon and rectum [7].

IBD is acknowledged as an immune-mediated disorder of the intestines, stemming from multifaceted interactions among genetics, environmental factors, and the gut microbiota [2]. According to studies conducted in the last few decades, several factors, including genetic transmission, intestinal immune disruption, gut microbiota disruption, diet, infection, lifestyle, psychological stress, sleep disorders, smoking, and early-life antibiotic exposure, can affect the development of IBD [8,9]. The exact underlying mechanism responsible for IBD’s pathophysiology remains poorly understood.

Immunologically, IBD is characterized by dysregulation of the mucosal immune system, involving the loss of immune tolerance and the emergence of uncontrolled immune responses to antigens derived from the normal gut microbiota [10]. More than 240 susceptibility loci were found by extensive genome-wide association studies, many of which involve genes encoding proteins that trigger adaptive and effector immune activities [11]. IBD patients deal with a heavy disease load, have difficulty completing everyday tasks, and run the risk of developing anxiety and depressive disorders [12].

### 1.2. Stress Definition

The definition of “stress” in medicine was initially provided more than 80 years ago by the Hungarian endocrinologist Hans Hugo Bruno Selye as the physiological adaptive reactions of organisms to harmful threats (stressors), whether endogenous or external, physical or psychological, or actual or perceived [13]. The stress system, which organisms have developed to preserve homeostasis under threat, incorporates physiological and behavioral adaptations via appropriate central and peripheral neuroendocrine responses and is incredibly complicated. The organisms may enter a condition known as cacostasis, in which many crucial physiological activities are compromised, and may develop various acute and chronic diseases when exposed to long-term or severe stress [8].

The flooding of adaptive capacities, which results in an excessive vital overflow beyond normal self-regulatory potential, is one of psychological stress’s principal impacts. Psychological stress is a specific kind of relationship between an individual and the environment [14]. The association between the stress factor and the emergence of several pathologies, including cardiovascular diseases [15,16], coronary disorders [16,17], high blood pressure [18,19], strokes [20], and brain damage [21,22], has been extensively researched by many different authors. Additionally, it is obvious that stress negatively affects a variety of illnesses [15,23]. Similar to other diseases, the research at hand appears to support the hypothesis that, among other things, excessive levels of stress have a role in the etiology, course, and responsiveness to the therapy of digestive problems [13,24]. In the current competitive environment, stress is unavoidable, and chronic stress is linked to negative consequences on physical health, including the development of IBD.

Stress is broadly described as a threat to a steady state of homeostasis in a person’s life, in contrast to psychiatric disorders like sadness and anxiety. It involves both a stressor (i.e., an environmental demand) and a person’s physiological and emotional reaction to the stressor [25]. The majority of Americans—60%—report experiencing stress in their daily lives [25]. In contrast, in a patient with anxiety, stress perception may be disproportionate to environmental demand. An exaggerated response to a stressor may be functionally disruptive and develop into a psychiatric disease. It is becoming more widely understood that significant psychiatric comorbidities have a negative impact on IBD patients’ disease activity and use of healthcare services, but it is less known how daily stressors and perceived stress affect patients’ IBD outcomes and disease courses [26]. Through its effects on the immunological, endocrine, and neurological systems, stress has been shown to negatively affect GI function and increase gut permeability.

## 2. Objectives of This Study

This review aims to assess the influence of psychological stress on the prevalence and outcomes of intestinal inflammation among pediatric and adult patients. The onset, degree of activity, and response to treatment among children and adults with IBD are incompletely understood and are still the subject of current research. There are a lot of factors influencing IBD activity (environmental, genetic factors, neuropsychological or intestinal microbiota imbalance, and intestinal immunity impairment). This paper aims to analyze the interplay between these factors and the course of the disease. Furthermore, this review sets out to quantify if a reduction in stressful factors decreases the incidence of IBD relapses and prolongs the remission time of the disease.

## 3. Materials and Methods

### 3.1. Search Strategy

The authors performed narrative literature research centered on the interaction between IBD and stress among adult and pediatric patients and also in animal subjects. This research was conducted by electronically searching PubMed, Scopus, Medline, and Cochrane Library databases from 2000 to December 2023.

### 3.2. Study Selection

All publications focusing on stress and pathogenic, clinic, and diagnostic aspects and therapeutic interventions in pediatric and adult patients with IBD were assessed. Also, in this narrative review, we included studies conducted on animals. The inclusion criteria used to extract relevant information included the following: clinical and preclinical/laboratory studies published in the English language in the last 23 years, with a sample size comprising more than 15 subjects. The most important data were summarized in this narrative review.

## 4. Stress and Gut Microbiota Brain Axis

The hypothalamic–pituitary–adrenal axis (HPA) represents the most important neuroendocrine system involved in the stress response of the host. The link between the perceived stressful stimuli and the physiological reaction to different types of stress is encoded by the HPA. The HPA, the autonomic nervous system (ANS), the CNS, the stress response, the GI corticotropin-releasing factor system (CRF), and the intestinal response (including the intestinal barrier, the luminal microbiota, and the intestinal immune response) are among the neural components that interact in the brain–gut axis [27].

The cholinergic anti-inflammatory system modulates the innate immunity action against different types of tissue alterations induced by infectious pathogens or hypoxia. It represents the efferent pathway of the inflammatory response, which is the neural feedback that regulates the inflammatory reaction. In the cholinergic anti-inflammatory system, through an anti-TNF-α, the action of the efferent vagus nerve may be a therapeutic target in IBD by a pharmacological, dietary, or neurostimulation strategy, according to animal studies [28,29]. The psychological requirements of patients with IBD are also highlighted by the psychophysiological susceptibility of these patients due to the potential presence of any mood disorders, distress, increased perceived stress, or maladaptive coping mechanisms. There is growing evidence that stress or other negative psychological traits may have an impact on the disease course; thus, clinicians need to talk to patients about these difficulties.

An organism’s reaction to a request from the environment is stress. Stress is a physiological response that can turn pathologic when there is an imbalance between the body’s ability to adapt and what it needs from its environment. This can cause functional, metabolic, and even lesion-related illnesses [30]. The typical mechanism by which stress results in an adaptation is the hypothalamic–pituitary–adrenal (HPA) axis. The main neuromediator of stress, CRF, when delivered directly into the brain, mimics the general endocrine, behavioral, autonomic, and visceral alterations brought on by stress in experimental animals [31].

Recent research [31,32,33,34] indicates the gut microbiota’s involvement in IBD. The neurological system and commensal, pathogenic, and probiotic microorganisms can communicate in both directions. Bacteria are capable of passing through the epithelial barrier during times of stress, triggering the mucosal immune response, and they can go to secondary lymphoid organs [32], activating the innate immune system. Mice exposed to a social stressor experience changes in the intestinal microbiota and an increase in the amount of cytokines in their bloodstreams; antibiotics counteract these outcomes [33]. Intestinal microbiota changes decrease resistance to intestinal pathogen-induced infection [34]. The results of these research studies demonstrate the method by which stress, the gut microbiota, and the immune response interrelate. The proliferation of bacteria is stimulated by the sympathetic nervous system’s release of catecholamines (norepinephrine) [35]. Stress-mediated changes might influence host susceptibility to infection and modify microbial colonization patterns on the mucosal surface. These modifications to host–microbe interactions could cause an impact on neuronal activity in stress-sensitive areas of the brain [36]. The microbiota brain–gut axis, which connects the gut and the brain, may be mediated by the intestinal bacteria. This is illustrated in Figure 1.

The ANS, alongside the HPA axis, modulates the efficiency with which the GI tract works. To control mucosal immune responses and other intestinal activities, such as nutrition absorption [37], the ANS is responsible for causing efferent signals to be transmitted from the CNS (brain and spinal cord) to the intestinal wall [38]. Afferent signals from the lumen of the intestine are also known to control behavior, sleep, and stress responses through enteric, spinal, and vagal nerve cells [39]. The enteric nervous system (ENS, the “second brain”), a component of the peripheral nervous system, primarily communicates with the CNS in a bidirectional pattern upon receiving inputs from the diet and gut bacteria [40]. The ENS can, however, also inherently innervate the gut in an autonomous way [41].

For the gut microbiota, the GI tract acts as a dynamic, local ecosystem. The gut microbiota is made up of around 35,000 different bacterial species, which are frequently divided into two major groups: Bacteroidetes and Firmicutes [42,43]. It is necessary for modulating processes associated with barrier function against pathogenic microorganism colonization, such as mucosal integrity [44], immunomodulation [45], and pathogen protection, in addition to its role in metabolism [46]. In recent times, preclinical, translational, and clinical studies [47,48] have indicated that modifications to the microbiome’s structural composition or function may play a crucial role in the occurrence of mental illness, including depressive-like behavior. According to some studies, the development of multifactorial chronic inflammatory illnesses, including IBD, has been strongly correlated with changes in the gut microbiota, indicating that dysbiosis is a significant component in both GI and mental health [49,50].

Short-chain fatty acids (SCFAs), such as butyric acid, propionic acid, and acetic acid, which are commonly observed to be reduced in mucosa and feces of individuals with IBD, are also produced by the gut microbiota through the fermentation of dietary fibers [51]. These metabolic products have been demonstrated to have a significant role in increasing epithelial cell proliferation [52], barrier function [53], and cellular metabolism, as thoroughly reviewed by Parada Venegas et al. [54,55]. By simulating G-protein coupled receptor signaling pathways, SCFAs have also been linked to the regulation of intestinal homeostasis and the prevention of pathogen colonization [56]. SCFAs are also known to have neuroprotective effects, which is pertinent. Gamma-aminobutyric acid, for instance, can influence behavior since it is an inhibitory neurotransmitter that has a role in anxiety and sadness [57]. Modifications in bacterial neurometabolites or bacterial cell wall carbohydrates are two more ways that the intestinal microbiota influences neuronal responses. These substances either directly affect primary afferent axons or cause the release of chemicals by epithelial cells that control neural signaling in the ENS [58].

Together, the complex interactions between the gut, microbiota, and brain enable intestinal and extraintestinal homeostasis, which regulates higher cognitive and affective processes as well as GI functions.

Early life stress can be related to admission to the Neonatal Intensive Care Unit (NICU). Many preterm infants are exposed to stress, pain, and complications of the GI system. One of the most dangerous GI neonatal emergencies is necrotizing enterocolitis (NEC), which is related to multiple risk factors. The most important is prematurity, which comes together with hypoxia, sepsis, abnormal colonization of the bowel, and the release of inflammatory mediators. These inflammatory mediators are set off by a triggering event linked to stress [59]. Early life stress can leave lasting effects. Some stressors may enhance growth and adaptation, but others may be innate to alter future health trajectories. For many preterm infants, early life exposure to persistent and intense stressors may become potentially toxic [60]. Recent studies have linked stress to the gut microbiome, leading to dysbiosis and suppressing the activation of the innate immune system in response to stress [61,62]. The gut colonization process is dynamic, depending on environmental factors. Preterm infants admitted to the NICU experience dysbiosis with an abundance of Gram-negative bacteria. There is a need to determine if stress is related to prematurity and antibiotic usage and, therefore, create an appropriate environment for bacteria abundance [60]. To optimize the management of NEC, it is necessary to identify critical diagnostic methods and their ability to determine the existence of future inflammatory bowel disease. Over the years, several studies have aimed to identify specific biomarkers [63,64,65,66]. A literature review was performed to update data on NEC biomarkers, which concluded that various proteins and products of metabolism can be used to determine NEC with modern technology. Nevertheless, future research is needed to determine non-invasive panels of high-value and diagnostic algorithms [67]. Long-term implications for NEC survivors include short bowel syndrome, cholestatic liver disease, and impaired neurodevelopment [68]. Since modern technology has arisen over the years, the link between NEC and the possibility of IBD development in the future has followed. Tremblay et al. used deep sequencing (RNA-Seq) to determine the gene expression profile in preterm infants diagnosed with NEC and non-NEC conditions. Analysis of the data indicated that the relevant functional pathways in preterm infants with NEC were associated with immune functions, such as altered T and B cell signaling, B cell development, and the role of pattern recognition receptors for bacteria and viruses. Genes strongly modulated in NEC neonates are significantly similar to those reported in CD, which is a chronic inflammatory bowel disease. The results of this cited study confirm that a large proportion of the significant functional pathways and phenotypes are common between NEC and CD and that some of the biomarkers used for diagnosing CD can be used for predicting NEC development in intensive care units [69]. This approach can be used as a point-of-care tool for diagnosing NEC or bowel inflammation, as demonstrated for lipocalin 2 and calprotectin [70,71,72].

Neonatal early-life stress is predominantly related to admission to the NICU. The NICU uses a patient-centered approach driven by protocols; notwithstanding, preterm newborns experience a significant level of stress from their surroundings since they must endure complex life-saving medical operations, protracted absence from their parents, and continuous and extreme stressors [60]. The vulnerability of newborns allows for the stress and complications associated with prematurity to also affect neurological development [73,74].

As stated before, the link between stress and gut microbiome has been proven. Questions have arisen regarding how these two can impact neurobehavioral development in preterm infants during NICU hospitalization. The gut microbiome is involved in the regulation of neurological, behavioral, and cognitive development [75,76,77]. Early alleviation and treatment of neurobehavioral abnormalities in preterm newborns can be made more accessible by identifying possible pathogens and understanding the pathogenic process of gut microbiota involved in neurobehavioral development.

Neurobehavioral outcomes are assessed using the NICU Network Neurobehavioral Scale (NNNS), which is a standardized score and includes, among others, stress, handling, and quality of movement [78]. It is shown that infants who had less acute stressful events during hospitalization had improved neurobehavioral outcomes [73,79]. Several gut bacteria were studied in terms of their link with compromised neurodevelopmental outcomes. Chen J et al. conducted a longitudinal study and identified eight gut microbiome bacterial communities associated with neurobehavioral profiles in early life. The vast amount of Enterobacteriaceae is linked to an increased NSTRESS score [79]. Another study suggested that Klebsiella overgrowth was associated with brain injury involving immunological alterations [75]. These findings suggest that targeted interventions positively impact developmental outcomes in infants.

The GI health of a newborn is an area of great interest; therefore, interventions are needed to establish a well-desired, healthy GI system. For digestion and nutrient absorption, newborns require the GI tract to mature structurally and functionally. A term newborn will have a normal neonatal adaptation with appropriate nutritional requirements. Preterm newborns are characterized by an immature GI system, which limits the utilization of enteral nutrition. In addition, factors such as stress, infections, and antibiotics contribute to impaired bowel function and gut–brain axis [80].

Antibiotic use in neonates admitted to the NICU is a common practice, which includes the most prescribed drugs in the NICU [81]. The gut microbiome is influenced by antibiotics and their associated side effects, which influence the early establishment of intestinal microflora [82,83]. The use of antibiotics within the first 2 weeks of life for preterm newborns is linked to an increased risk of late-onset sepsis, NEC, or death [84]. The randomized study REASON [85] was conducted to determine if antibiotics should be used in the first 48 h after birth and their effects on gut microbiome and inflammatory status. The results suggest that using antibiotics in the first 48 h of life has no long-term effect on the microbiome. Moreover, the microbiome diversity is recoverable. The REASON study also suggests that Bifidobacteria may influence GABA signaling in the brain. Thus, antibiotic use can interfere with the gut–brain axis [85].

Postpartum Gi colonization is influenced by additional factors such as delivery mode and feeding regime. Newborns delivered by cesarean section have a reduced diversity of gut microbiome compared with those vaginally delivered [86,87]. Delivery mode has long-term effects on dysbiosis, leading to autoimmune and metabolic disorders [88]. A factor of debate in neonatal care is the use of prebiotics and probiotics to regulate the microbiome. Dermyshi et al. conducted a systematic review and concluded that probiotic use was significant in preventing severe NEC, late-onset sepsis, and all-cause death in infants with very low birth weight [89]. There are some arguments against probiotic use based on knowledge of some conditions of preterm newborns, such as extremely low birth weight, the immaturity of the immune system, susceptibility to infections, and similarly, the conjecture of probiotic formulas and doses [90,91]. Bifidobacterium species are major colonizers of the infant gut, comprising about 70% of the gut microbial population while breastfeeding [92]. These species have become the standard bearers for probiotic formulations, considering their unique abilities to metabolize complex carbohydrates from human milk—human milk oligosaccharides (HMOs) [93]. A recently published review highlights that infants treated with probiotics have bountiful Bifidobacterium spp. independent of the probiotic formulation and reduction in potentially pathogenic bacteria [94]. Probiotics containing both Bifidobacterium and Lactobacillaceae can influence the preterm gut microbiome configuration while promoting the development of a microbiome that is more typical of term newborns [95].

After delivery, the gut microbiome is influenced by interactions between the mother, the newborn, and environmental factors. The feeding regime is one essential variable. Breastmilk is the optimal nutrition for a newborn. Nevertheless, not all newborns can be fed with their own mother’s milk. Cesarean section can delay the breastfeeding process [96,97]. In addition, the immature GI system of preterm newborns will delay the feeding process. Human milk composition influences intestinal immunological processes and digestion, including GI colonization. Human milk microbiota is the second source of microorganisms for an infant. Thus, early gut colonization with human milk is essential for developing the immune system [98]. Breast milk is to be recognized as the gold standard for neonatal nutrition. However, in some circumstances, breast milk is unavailable, and formula feeding is required. The modernization of milk formula is a continuous process to create the best alternative regime [99]. HMOs are now artificially synthesized to be added to milk formula. These complex carbohydrates abound in human milk, modulate microbial composition, impede pathogenic invasion, and influence the immune response [100]. The benefits of human milk extend to premature newborns through its potent trophic effect on the immature gut. This valuable effect leads to earlier full feeding and prevention of late-onset sepsis and NEC. A good feeding practice is one key component to decreasing the burden of prematurity [101]. The GI health of a newborn plays a crucial role in the short term as well as in the long term. As discussed, there are some potential interventions to alleviate the effects of early life stress on the GI environment. Notwithstanding, further research is needed to provide a relevant understanding of the complex interplay between early life stress and neonatal outcomes.

Early childhood has a relatively complex and unstable gut microbiota; thus, any change is likely to influence the intestinal immune system and predispose people to IBD. Antibiotics, birth control, and non-steroidal anti-inflammatory drugs (NSAIDs), for example, have been shown to increase the risk of IBD, most likely by changing the commensal flora and/or intestinal barrier [9]. More specifically, a meta-analysis revealed that antibiotics were more strongly linked to an elevated risk of newly developing CD than UC [102]. Accordingly, a search across numerous databases showed that, compared with people who were not exposed to the medicine, those who used oral contraceptives had a 24% and 30% higher chance of getting CD and UC, respectively [103]. Similarly, long-term use of large dosages of NSAIDs [104] led to the worsening of IBD [105], possibly by non-selective inhibition of the cyclo-oxygenase [106].

Table 1 presents the studies conducted on humans and animals that investigated the link between stress and gut microbiota.

As shown by these studies, appropriate physiological responses to stress and/or immunity are necessary for survival. As such, aberrant responsiveness can be detrimental to the host, leading to the development of chronic disorders, including IBD [132] and brain disorders [135].

## 5. Stress-Induced Alterations/Inflammation in the Gastrointestinal Mucosa

In individuals with inactive IBD, unfavorable life events, depression, and chronic stress increase the risk of relapse, according to recent research. It has long been suggested that psychological stress enhances disease activity in IBD. According to recent studies, stress-related changes in GI inflammation may be mediated by altered bacterial–mucosal interactions, altered HPA axis function, mucosal mast cells, and mediators like CRF.

The hypothalamus, amygdala, and hippocampus are three particularly intertwined brain areas involved in the complex integration of the stress response. Higher cortical structures as well as visceral and somatic afferents provide input to this network. The HPA axis and the ANS are two interrelated effector pathways by which they control the neuroendocrine stress response. Adrenocorticotrophic hormone (ACTH) is released from the anterior pituitary gland in response to the stimulation of CRF release from the hypothalamus by stress. Cortisol, the main glucocorticoid, is then secreted from the adrenal cortex as a result [136,137].

The hypothalamus’s direct descending neuronal routes to the pontomedullary nuclei, which regulate the autonomic response, are activated by stress. The adrenal medulla releases adrenaline and noradrenaline in reaction to stress by stimulating the sympathetic nervous system. The vagus and sacral nerves offer parasympathetic input to the upper gut and the distal colon and rectum, respectively, while sympathetic neurons of the sympathetic ANS also directly supply the entire gut. The ENS, the gut’s abundant nerve supply, and the efferent and afferent neurons of the sympathetic and parasympathetic ANSs communicate with one another. This network is known as the brain–gut axis. The 100 million neurons that make up the ENS control the GI tract’s motility, exocrine, endocrine, and microcirculation functions [136,138].

The lymph nodes, the mucosa-associated lymphoid tissue, bone marrow, spleen, and thymus all have intimate effector junctions formed by nerve fibers of the ANS with lymphocytes and macrophages. At the neuron–immune cell junction, some neurotransmitters, contained in the neurons of the ENS and ANS, including catecholamines, vasoactive intestinal peptide, angiotensin II, neurotensin, somatostatin, and substance P (SP), can affect lymphocytes, macrophages, neutrophils, and other inflammatory cells [136,138].

Glucocorticoids, which are released from the adrenal cortex in response to ACTH from the pituitary gland, primarily depress the immune system when present in high amounts. They promote the production of anti-inflammatory proteins like IL-10, IL-1 receptor antagonists, and lipocortin 1 [139]. Glucocorticoids have an inhibitory effect on the transcription factors AP-1 and nuclear factor-kB, which in turn inhibits the production of several inflammatory chemokines and cytokines, including TNF-α, IL-1, and IL-6. T cells and eosinophils are two examples of inflammatory cell types that glucocorticoids also encourage to undergo apoptosis. On the other hand, cortisol has an immunostimulatory impact in lower amounts [138,140,141,142].

The complexity of psychological stress’s effects on the body’s immune and inflammatory systems depends on the stressor’s length and degree. Affected systemic immunological and inflammatory function is linked to both acute and chronic stress, which may be relevant to the pathophysiology of IBD [143].

Chronic persistent stress, such as that brought on by unfavorable life circumstances, results in a protracted rise in cortisol over several days, which is typically accompanied by immunosuppression. Reductions in macrophages, CD8+ lymphocytes, and NK cells have all been linked to divorce, bereavement, and depression [143,144,145,146]. Chronic psychological stress, however, has also been linked to subclinical increases in inflammation, in addition to immunosuppression. Serum C reactive protein levels have been reported to be elevated in individuals with depression as well as middle-aged and elderly patients with lower heart rate variability, a marker of chronic stress, and increased sympathetic tone [147,148].

Experimental stress tests and real acute stress both stimulate the sympathetic nervous system acutely and result in an almost immediate rise in adrenaline and noradrenaline levels. Cortisol levels then increase, but these shifts only last for a short time. This method of stimulating the stress axes has been linked to improved immunity. Inflammatory cytokines, which are known to play a significant role in the pathophysiology of IBD, are produced by whole blood in greater quantities and at higher serum levels. Additionally, it has been demonstrated that acute stress can quickly redistribute the lymphocyte population and result in leucocytosis in both healthy individuals and people with UC in the remission phase. Also, there is an increase in the proportion of NK cells and CD8+ cytotoxic T cells and a commensurate rise in their cytolytic activity [136,143,149].

Experimental stress has been demonstrated to promote platelet activation, as measured by aggregation and production of inflammatory mediators and platelet-dependent thrombin generation in healthy participants, as well as in patients with UC in remission. Patients with IBD have increased levels of platelet activation in their blood, which may play a role in pathogenesis by promoting the production of thrombi and microinfarcts as a result of microvascular ischemia. Instead of aspirin, beta-blockers may be able to suppress the stress-induced activation of platelets, indicating that sympathetic stimulation is an important factor in the process. Acute experimental psychological stress also causes platelet–leucocyte aggregation formation; this factor is elevated in IBD patients and may make it easier for leucocytes to extravasate to specific regions of inflammation [136,150,151,152].

It has been suggested that abnormal mucosal immunity to the intestinal microbiota is the cause of IBD. The growth and maintenance of secretory cells depend on the transcription factor X-box-binding protein-1 (XBP1), which is associated with JNK activation and an essential part of the endoplasmic reticulum (ER) stress response.

It was demonstrated that XBP1 deletion in intestinal epithelial cells (IECs) causes spontaneous enteritis and enhanced vulnerability to produce colitis as a result of both Paneth cell insufficiency and IEC hyperactive responses to the IBD-inducing agents TNF-α and flagellin [153,154]. As a result, intestinal inflammation can only result from XBP1 aberrations in IEC, establishing a link between the development of organ-specific inflammation and cell-specific ER stress.

Inflammation may be brought on by a stressful external setting in cells with high secretory activity. If true, cell-specific XBP1 deletion that induces ER stress in vivo may result in organ-specific inflammation and offer a molecular explanation for the onset of proinflammatory illnesses. The intestinal epithelium contains four highly secretory cell lineages that are exposed to high concentrations of exogenous antigens: goblet and Paneth cells, absorptive epithelium, and enteroendocrine cells, that descended from a common, continuously regenerating intestinal epithelial stem cell [154,155].

The studies showed that spontaneous enteritis results from ER stress induction in intestinal epithelium caused by tissue (and cell type)-specific disruption of XBP1 because XBP1-deficient IECs are unable to generate antimicrobial activity and respond suitably to inflammatory signals in the local environment. The XBP1 gene locus on chromosome 22q12.1 contains some single nucleotide polymorphisms that increase the risk for CD and UC, establishing the ER stress pathway as a common genetic contribution to IBD [154,155].

## 6. The Role of the Nervous System as an Immune Modulator in Patients with IBD

The brain–gut axis is an intricate, bidirectional system that includes several connections between the GI tract, the autonomous nervous system, and neuroendocrine pathways. This network may have an impact on the emergence of functional GI disorders like irritable bowel syndrome (IBS), gastroesophageal reflux disease, and IBD [8,156,157].

The pathogenesis of IBD is thought to be influenced by psycho–neuro–endocrine–immune regulation via the brain–gut axis. The stress system (the HPA axis), the ANS, the CNS, the (GI) CRF system, and the intestinal response (which includes the intestinal barrier, luminal microbiota, and the intestinal immune response) are among the neural components that interact to form the brain–gut axis [157,158].

It is crucial to investigate both the drivers of the immunoinflammatory response as well as the peripheral mediators of inflammation (cellular components and their byproducts) to comprehend the etiology of IBD. For IBD, the gut microbiome has recently drawn more attention as an essential component of this process. To date, IBD therapy has focused on treating the phenotypic manifestation of IBD; however, psycho-neuroimmune modulation may be the platform that connects the human experience, mental state, gut microbiome, and immune response [158,159].

It is known that the brain system regulates immune function, and this ability could be used to inhibit the immune system in IBD. Since vagus nerve stimulation (VNS) was demonstrated to diminish local and systemic inflammation in animal models of endotoxemia, arthritis, and colitis, the vagus nerve and its primary neurotransmitter acetylcholine have received particular attention [160,161,162,163,164].

VNS decreases inflammation, but the exact mechanism by which it does so is still being investigated. It is unclear if the vagal efferent nerves genuinely innervate mucosal cells, despite the intestine being highly innervated. This might point to a function for different kinds of nerves. It has also been shown that a wide range of neurotransmitters, including ACh, as well as nitric oxide, adrenaline, norepinephrine (NE), and a huge number of neuropeptides that function as immune modulators, may act directly on a variety of immune cells in the gut [162,163,164].

The autonomic nerve system, which operates autonomously in that its actions are not directly under conscious control, regulates important digestive tract functions like motility, secretion, and vasoregulation. Based on anatomy and neurotransmitter activity, it is categorized as sympathetic and parasympathetic branches and reflects the extrinsic regulation of the gut. The intrinsic neurons of the ENS are found within the wall of the GI tract, and the sympathetic and parasympathetic systems have their origins in the CNS (with cell bodies in the brainstem and spinal cord). The ENS is a unique component of the CNS that can function on its own or in reaction to external signals coming from sympathetic and parasympathetic nerves. The enteric ganglia, which are clusters of nerve cells that form the ENS, generate nerve fibers that innervate effector tissues like the gastroenteropancreatic endocrine cells, blood vessels, and the intestine’s muscular layer. The ENS comprises the myenteric plexus and the submucosal plexus [160].

ACh is one of the neurotransmitters found in the ENS, which is thought to control gut immunity. Neuropeptides, acting as crucial mediators between the nervous system and neurons or other cell types in the effector tissues, are also a part of the crosstalk between the gut and nervous system. These tiny proteins play a crucial role in multimodal neural communication, including SP, vasoactive intestinal polypeptide, calcitonin-gene-related peptide, neuropeptide Y, somatostatin, serotonin, and CRF.

When the cholinergic anti-inflammatory pathway was originally discovered, the impact of neurons on gut inflammation first came to light. According to one theory, the neurological system uses this pathway as a reflex mechanism to regulate abnormal, increased immune responses [165,166].

Since inflammation of the gut also affects the nerves and their adrenergic activity, there is a reciprocal relationship between the SNS and the inflamed intestine. In the inflamed colonic mucosa of IBD patients as well as in several colitis animal (mouse) models, sympathetic innervation is noticeably reduced. A loss of tyrosine hydroxylase (TH) nerve fiber was shown in patients with IBD, where TH is the rate-limiting enzyme for the synthesis of epinephrine and NE. Additionally, it was discovered that proinflammatory SP+ fibers predominated noticeably [167].

It makes sense that a decrease in sympathetic neurotransmitter levels would follow the loss of sympathetic nerves in inflamed tissue. CD patients had markedly lower NE levels than healthy controls. This is corroborated by the observation that in inflamed tissue, the release of NE from sympathetic nerve terminals is constrained. As NE-negative immune regulation is reduced, inflammation-induced inhibition may increase the chronicity of the inflammation. Furthermore, the primary function of sympathetic nerves is vasoregulation, which is distinct from the SNS’s anti-inflammatory function [168,169].

When these nerves are lost, blood flow is hampered, which may contribute to maintaining the inflammatory environment. According to the concentration of neurotransmitters and neuropeptides (which is dependent on their release and the presence of sympathetic nerves), the quantity and accessibility of receptors, the receptor affinity, and the timing of sympathetic activity, the SNS has opposing proinflammatory and anti-inflammatory functions. There is disagreement over how the SNS and the inflammatory environment contribute to the persistence of inflammatory processes [168,169]

## 7. Current Publications on the Interplay between Stress and Psychosocial Disorders in Children and Adults with IBD

Previously conducted systematic reviews have confirmed that individuals diagnosed with IBD exhibit a decreased quality of life in comparison with the general population [170]. This decline in quality of life is more pronounced during active phases of IBD rather than inactive periods, and it is particularly notable in individuals with CD as opposed to UC [171]. However, the quality of life tends to improve as the disease progresses over time. Anxiety and depression commonly co-occur as comorbidities in individuals with IBD and have a reciprocal relationship with the disease [172]. Nevertheless, the precise nature of this relationship between anxiety, depression, and IBD has yet to be established due to the limited availability of prospective study designs [173]. A recent meta-analysis investigating the potential causal link between anxiety, depression, and the exacerbation of symptoms in IBD yielded inconclusive findings [174]. Psychological interventions have only demonstrated modest beneficial effects on the quality of life and depression experienced by individuals with IBD, and further clinical trials are necessary to ascertain their impact on disease activity [175]. Consequently, there remains a critical need for an enhanced understanding of the psychological factors influencing individuals diagnosed with IBD.

According to Lazarus and Folkman, stress is defined as a complex interaction between an individual and their environment, where the individual perceives the demands and challenges as overwhelming, surpassing their available resources, and posing a threat to their overall well-being [176]. This relationship is influenced by various factors, including personal and environmental factors, the nature of the stressor itself, how it is perceived and evaluated, and its immediate and long-term effects. Emotions play a significant role in the appraisal of stress, as recognized in both psychological [177] and physiological [178] models, leading some researchers to use the terms stress and distress interchangeably.

Stress is believed to activate the HPA axis, which has been implicated in the inflammation of the GI system [136]. Changes in this axis, such as alterations in the neuroendocrine-immune system, can contribute to the likelihood of disease exacerbation [179]. Additionally, stress can lead to behavioral changes, including nonadherence to medication [180], poor dietary choices [181], and alcohol consumption [182], all of which increase the risk of disease flare-ups. Measures of perceived stress may be valuable in identifying individuals at risk of relapse and indicating the need for intervention when used alongside disease activity indices.

## 8. Anxiety and Depression in Children and Adults with IBD

Psychiatric disorders have a significant negative impact on various aspects of IBD. Depression has been linked to an elevated risk of disease relapse and poorer treatment response [183,184,185,186]. Similarly, anxiety is associated with a higher likelihood of undergoing surgery, lower adherence to medication, and a diminished quality of life [183,187,188]. In a comprehensive study involving multiple institutions, after accounting for confounding factors, the presence of comorbid depression and/or anxiety was associated with a 28% increased risk of surgery in individuals with CD. Moreover, it was correlated with a higher number of colonoscopies and an increased probability of utilizing immunomodulators as part of the treatment regimen [189].

Although psychiatric disorders have been observed to manifest after an individual is diagnosed with IBD [190], there is evidence suggesting that these disorders may predate the diagnosis of IBD by several years [191,192]. Most studies investigating this relationship have primarily focused on depression or anxiety, placing greater emphasis on the prevalence rather than the incidence of psychiatric disorders. However, understanding the incidence of these disorders is crucial for unraveling their etiology. Considering the detrimental impact of comorbid psychiatric disorders on the progression of IBD, it is vital to thoroughly examine the burden of psychiatric comorbidity in individuals with IBD.

The highest occurrence of CD is typically observed during the third decade of life, while UC incidence begins to rise in the same decade and maintains more consistent incidence rates across different age groups [193]. However, IBD typically affects individuals during the crucial period of social and career development [194]. With an increased occurrence of psychiatric disorders following an IBD diagnosis, there is potential for an enduring burden of mental health issues in individuals with IBD. Limited research has been conducted specifically on the management of mental health disorders in individuals with IBD [195]. Considering the potential significance of the brain–gut axis in the pathobiology of IBD [157], it is possible that the inflammatory state of the gut can influence brain function and mental well-being. Therefore, assuming that treatments for psychiatric disorders in the general population will be equally effective for individuals with IBD would be an oversimplification. While the mental health effects of corticosteroids are well-known, there is limited information regarding the potential harms or benefits of mental health associated with various effective biological therapies used for IBD. Consequently, there is a need for extensive research to optimize therapy for individuals with IBD who develop a psychiatric disorder.

## 9. Mendelian Randomization Studies Evaluating the Causal Associations between IBD and Psychological Conditions

Compared with the general population’s prevalence (3.4%), anxiety is a common comorbidity in people with IBD, ranging from 19.1% to 35.1% [172,196,197]. Because of the potential pathophysiological pathways mediated by the gut–brain axis, there has been a lot of interest in the relationship between anxiety and IBD, or vice versa.

A few observational studies have looked into the temporal relationship between anxiety and IBD and theorized a bidirectional relationship between the two conditions. Anxiety may be more common in IBD patients than in people who only seek medical check-ups [198,199,200].

Also, throughout a 10-year follow-up period, newly diagnosed IBD patients experienced an increased prevalence of anxiety (incidence rate ratio: 1.39) in comparison with matched control persons [199]. Observational studies have demonstrated that those with anxiety are more likely to have IBD, but the reason is unknown [200]. During 6.7 years of follow-up, cohort research revealed a greater prevalence of IBD in patients with newly diagnosed anxiety than in control persons [201].

Using bidirectional Mendelian randomization methodology, He Y and collaborators conducted a new study to examine the causal connection between IBD and anxiety. MR is a genetic technique that determines the causal relationship between an exposure and an outcome via genetic variants found in genome-wide association studies, typically utilizing single nucleotide polymorphisms (SNPs) [196]. The findings indicated that a higher incidence of anxiety could be related to a genetic vulnerability to UC (odds ratio: 1.071 (95% confidence interval: 1.009–1.135), *p* = 0.023). However, anxiety was not related to a genetic predisposition to CD. But neither UC nor CD have been related to a genetic predisposition to anxiety. This cited study demonstrated the strong relationship between anxiety and genetic susceptibility to UC, emphasizing the value of early screening and effective treatment for anxiety in UC patients [196].

It is still unclear how exactly UC or CD causes anxiety from a physiologic standpoint. Some research suggests that the inflammation-regulated gut–brain axis can impact neuronal development and ensuing behavioral traits [156,202].

The blood–brain barrier allows for circulating leukocytes and cytokines to enter the brain, where they can cause neuropsychiatric diseases [156]. The exact biochemical mechanisms by which UC influences the development of anxiety are still unknown, even though He Y and collaborators’ investigation examined the causal association between IBD and anxiety [196]. For example, it is unclear how the gut–brain axis plays a part in this process. Therefore, to fully understand the molecular mechanisms, additional basic and clinical research is required to identify important regulators and pathways.

## 10. Quality of Life among Pediatric and Adult Patients with IBD

IBD significantly impacts quality of life and comes with personal, emotional, and social burdens. Numerous studies have shown that health-related quality of life (HRQOL) is impaired in patients with IBD compared with the general population. While disease activity and severity impact physical and psychological HRQOL, patients may also suffer from psychological problems during clinical remission. Impaired quality of life can affect the employment, family planning, and personal life goals of people with IBD. Improving quality of life requires a multidisciplinary approach that, together with adaptive coping mechanisms, helps to manage illness perceptions and reduce psychosocial burden. HRQOL is a concept that encloses those aspects of overall quality of life that have a demonstrable impact on physical or mental health. Measuring HRQOL can help improve connections between medical management and personal life and guide strategic plans [203]. Measuring and assessing the disabilities associated with IBD and the impact on the quality of life of those affected is critical to understanding the often-hidden burden that this disease places on those affected and society. This section of this review describes the impact of IBD on patients’ quality of life.

Several studies showed that persons diagnosed with IBD have significantly lower HRQOL in comparison with the general population, with an emphasis on severe disease activity [204,205,206,207,208]. Wilburn et al. conducted a study focused on patients’ appraisal of the disease rather than symptoms and medical management. It emphasized that IBD affects the daily routine and the need for fulfillment–nutrition, hygiene, self-esteem, attractiveness, and intimacy. The respondents indicated that their diets were restricted, and small portions replaced large meals. They reported hygiene concerns. It was common for them to carry deodorants, spare clothing, and wet wipes. Suffering from the disease meant they had to orient their lives according to the location of toilets. Finding the nearest public toilet became the first task when arriving at an unfamiliar location. Respondents reported being cautious about getting emotionally close to people because they found it difficult to initiate relationships [209]. IBD influences the need for fulfillment by affecting self-image, attitude toward life, and well-being [210]. This includes traveling as well. Foreign travel for people with IBD increases the risk of morbidity due to exacerbations, infectious diseases, and a lack of healthcare providers abroad. It is recommended that travelers seek thorough pre-trip counseling and vaccinations to ensure they are equipped with the appropriate information and resources to stay healthy during their journey [211,212]. A diagnosis of IBD limits travel for many patients; however, they should be counseled about travel behavior and seek medical advice before traveling abroad [213].

IBD is a chronic disease that alters individuals’ quality of life due to symptoms, medical management, depreciation of body image, psychological disorders, and long-term treatment. The sexual functioning of patients with IBD is a constant concern. The first report of sexual dysfunction in IBD dates back to 1978, which reported less sexual intercourse or even ceasing sexual intercourse [214]. Since then, several studies have emphasized that sexually dysfunctional rates are higher in patients with IBD than in the general population, affecting women more than men [215,216]. A meta-analysis regarding sexual dysfunction in IBD was performed in 2019, which confirmed that prevalence is higher in patients with IBD, reporting a relative risk of 1.41 for SD in men and 1.76 for SD in women [217]. Sexual dysfunction is a common yet little-known problem in patients with IBD. The control of IBD activity, together with mental and sexual health, is relevant to the well-being of individuals with IBD. For these patients, it is recommended to perform an early investigation of sexual dysfunction to improve their quality of life [218].

HRQOL is a key indicator of adjustment in adolescents with chronic diseases and has been used as an outcome in clinical trials and as a marker of treatment efficacy. HRQOL in adolescents with IBD has recently received considerable attention, given the physical and emotional demands of treating the disease. The treatment of IBD can include dietary changes, medication management, and surgical interventions [219]. Although medical interventions can relieve disease-related symptoms, they can also lead to adverse side effects impacting an individual’s HRQOL. In addition, the treatment of IBD and its symptoms can lead to increased self-consciousness and affect school and social life [220,221]. Youth with IBD are at risk for lower HRQOL than typically developing youth, including a higher risk for psychosocial, physical, and academic impairments. Kunz et al. conducted a study that compared youth and parent-proxy reports of HRQOL among youth with IBD to published comparison group data. It emphasized that youth with IBD reported lower psychosocial functioning than the healthy comparison group, higher physical and social functioning than the chronically ill group, and lower school functioning than all published comparison groups. Specific factors of IBD that may interfere with school functioning include frequent use of the bathroom, limited participation in physical classes, and embarrassing cosmetic side effects associated with having the disease or taking certain medications. Interestingly, even though more than half of this sample had an inactive illness, adolescents still reported deficits in academic performance compared with adolescents with an acute illness, suggesting that impairments in academic performance may persist even after symptoms of the illness have resolved. Even when the illness is dormant, the social stigma associated with the disease is perpetuated by frequent toilet use, visits to the nurse, and the need to take medication during the school day [222]. In addition, family stress levels can also affect the HRQOL of the youth. The study conducted by Gray et al. showed that high levels of parental stress were associated with lower HRQOL among adolescents with IBD [223]. The burden of IBD also takes its toll on the family. Knez et al. reported in their study that parents of children with IBD reported lower psychological and physical health than parents of healthy children [224].

Understanding how IBD impacts an individual’s day-to-day life means defining disability as a limitation in the ability to engage in usual activities. Work capacity was the most common IBD-related criterion for disability. However, more than this metric is needed because it does not capture all important aspects of the burden of this disease. In addition, in a relapsing–remitting disease such as IBD, the inability to work may be temporary and difficult to resolve. Although studies have demonstrated increased unemployment, sick leave, and disability retirement in IBD patients, most patients can work for many years after a diagnosis of IBD. They have lower productivity and fewer working hours than healthy controls, resulting in economic losses to individuals and society. However, the limitations are building interpersonal relationships, life activities, and mental well-being [225,226].

## 11. Lifestyle Factors and Psychological Stress in IBD Patients during the Coronavirus Pandemic Period

The coronavirus disease (COVID-19) is an infectious disease caused by severe acute respiratory syndrome Coronavirus-2, first reported in Wuhan, China, and soon spreading worldwide, affecting millions of people. Therefore, the interest of this review was also to examine the effect of the COVID-19 pandemic on lifestyle factors and psychological stress in patients with IBD.

For instance, Yu Nishida et al. performed a retrospective study of patients with IBD that compared the lives of 451 patients with UC or CD before and during lockdown. As illustrated in Figure 2, some aspects of their lives were affected. It appears that exercise time, walking time, and working time decreased while sleep time increased. The pandemic has created, among many other problems, anxiety, depression, and stress. In terms of psychological stress, IBD patients experienced a significant increase in stress from being at home and from being unable to exercise, while IBD itself remained equally stressful [227].

Regarding age, elderly patients, compared with younger, patients experienced less stress during the pandemic, and in terms of gender differences, men were found to be more affected [227,228]. Moreover, the pandemic has affected eating habits, leading to involuntary weight gain in both children and adults with IBD [228]. 

As a chronic and debilitating disease, IBD requires regular check-ups, but during the pandemic, hospitals were ordered to care for patients with COVID-19 rather than IBD patients [228], yet treatment adherence did not decrease [227]. On the other hand, limited access to medication and specialized treatment led to an increased risk of exacerbations [229].

Moreover, Conti C et al. conducted a cross-sectional study to determine the extent to which the quality of life of IBD patients was affected. The study included two groups of IBD patients, one recruited before the pandemic and the other during the COVID-19 outbreak, with a total of 902 people enrolled. Disease activity, somatization, and quality of life were investigated. Almost half of the patients reported symptoms, predominantly UC patients, and, as expected, a higher level of anxiety, depression, somatization, and implicitly a lower quality of life [229].

The fear of being admitted to a hospital during the pandemic has led to delays in the diagnosis and treatment of IBD flares, thereby increasing the need for urgent surgery and the risk of medical treatment failure [230].

The prevalence of COVID-19 infection in patients with IBD appears to be lower than in the general population [231]. Children with IBD are less affected by COVID-19 than adults with IBD, but they play a significant role in the spread of the virus. Therefore, strict hand hygiene remains essential to prevent infection in any category of patients [232].

Regarding IBD patients infected with COVID-19, a study by Sansotta N. et al. in Lombardy, the region of Italy that was most affected by COVID-19, points out that children with IBD under immunosuppressive therapy are not at greater risk of contracting the virus. Even if they do become infected, the symptoms are mild, so they do not require treatment adjustment or hospitalization to control IBD. However, IBD patients with comorbidities or treated with corticosteroids may develop a severe form of COVID-19. Still, no concrete evidence supports a direct link between them [233]. 

A study by Turner et al., using the Surveillance Epidemiology of Coronavirus Under Research Exclusion for Inflammatory Bowel Disease (SECURE-IBD), examined the first eight pediatric patients with IBD and COVID-19. These patients showed only mild symptoms such as fatigue, low-grade fever, and cough. Despite undergoing immunomodulatory therapy, none of the patients required hospital admission, and there were no reported deaths [234]. 

A group of 209 children from 23 countries with IBD who contracted COVID-19 was analyzed by Brenner J. et al. According to the study, only 7% of the children required hospitalization, which was due to a combination of pre-existing medical conditions, active IBD disease, GI symptoms, or the use of certain medications such as sulfasalazine/mesalamine and steroids. Two children developed a secondary infection and multisystem inflammatory syndrome and were admitted to the pediatric intensive care unit (PICU), where they required mechanical respiratory support. However, with appropriate treatment, the prognosis was good. Moreover, the study found that TNF-α was linked to a lower risk of hospitalization in these children [235].

Patients undergoing IBD treatment may experience lowered immunity as a side effect. This has raised concerns about whether the treatment should be modified. However, studies have shown that despite the increased risk of infections caused by lowered immunity, it is not recommended to interrupt treatment because of the high risk of disease activation [232,233,234]. Additionally, biological treatment may help prevent the cytokine storm associated with SARS-CoV-2, thereby having a protective effect and reducing the likelihood of patients contracting the disease by up to 5 times [236]. Nevertheless, some precautions should be taken with corticosteroid therapy. It is recommended to avoid starting a new steroid therapy and to decrease the dose of prednisone if it exceeds 20 mg/day. In children with CD, thiopurines may be used as an alternative to corticosteroids. Still, they are not without adverse effects as they increase the risk of viral infections including varicella-zoster virus, cytomegalovirus, and Epstein–Barr virus [231,237].

According to the British Society of Gastroenterology, vaccination against COVID-19 is safe, although there is a possibility of a lower immunological response in patients undergoing immunosuppressive therapy [231].

The negative effects of the pandemic on patients with IBD persisted even after its end. This condition is known as long COVID and is recognized as a health issue with a detrimental impact on patients. The most prevalent symptom is asthenia, and women seem to be more susceptible to this condition [238].

Therefore, the COVID-19 pandemic has certainly been a burden on the healthcare system, with still important consequences, especially for patients with IBD.

## 12. Association between Psychological Stress and IBD Outcomes/Relapses

Several studies have used their resources to examine the relationship between psychological stress and IBD outcomes/relapses. It turns out that stress is a trigger of the HPA axis, which, as mentioned before, affects the GI tract.

A systematic review by Black J. et al. demonstrates a link between stress and IBD disease activity. Although different subtypes of stress have been analyzed, perceived stress seems to be the key factor involved in exacerbating IBD. Therefore, measuring perceived stress using the Perceived Stress Scale as an assessment tool may predict an exacerbation episode [239].

A prospective population-based study reported that a major stressful event can cause an activation of the disease in the following 3 months. The most stressful situations turned out to be related to family, followed by work, school, and financial stress. It also established a strong inverse association between perceived stress and disease outcome [240]. Regarding UC, it has been observed that long-term perceived stress can triple the risk of UC flares [241].

Moreover, three prospective observational studies claimed that there is a psychological basis behind the course of the disease. This argument was supported by the fact that patients with UC in clinical and endoscopic remission, following repeated stressful events, triggered a relapse of the disease [242]. For instance, Jennifer L et al. showed that a better psychological adjustment can reduce perceived stress and therefore decrease the number of hospitalizations due to IBD flares [243].

Furthermore, Bonaz and Bernstein strongly suggested that there is a bidirectional relationship between active disease and stress, as being stressed can trigger symptomatic disease, and being symptomatic can exacerbate or even encourage the state of stress [157].

In addition, Sunavcky A. et al. aimed to study the psychosocial mediators underlying the relationship between illness severity and perceived stress. Thus, as shown in Figure 3, four psychosocial variables (catastrophizing, illness stigma, illness uncertainty, and illness shame) were investigated. Except for illness stigma, all the other variables were equally strong positive mediators between IBD severity and perceived stress [244].

Chronic psychological stress accelerates the progression of IBD and also leads to behavioral comorbidities such as anxiety and depression [245]. It is known that quality of life is affected in patients suffering from a chronic disease and that stress can even cause behavioral changes such as alcohol consumption, non-adherence to treatment, or a poor diet, thereby increasing the risk of IBD relapses [239].

Therefore, high levels of stress and its effect on IBD progression is still a topic of great interest, as it is difficult to determine whether psychological stress is a predisposing and contributing risk factor to the disease or the result of a chronic debilitating disorder such as IBD [246].

## 13. Interventions That Contribute to Stress Reduction in IBD

IBD is a chronic, relapsing pathology of unknown etiology, which carries the burden of affecting millions of people worldwide [247]. The pathophysiology of IBD is multifactorial, relying on an interaction between genetic factors, the microbiome, the immune system, gut mucosal integrity, and environmental triggers. The link between the immune system, nervous system, and psychological processes plays an essential role in IBD. Psychological stressors affect the gut through the increased production of proinflammatory cytokines, activation of macrophages, and TNF-α via the HPA axis [181,248]. IBD and psychological disorders share common proinflammatory pathways, which explains the association between flare-ups of IBD in individuals with depression and the poorer outcomes of IBD patients diagnosed with mental illnesses [249]. In addition to this is the gut–brain axis, defined as a bidirectional network between the nervous system and the intestines. The main component of this axis is the ANS, in which the vagal nerve plays the main role. It is well known that stress inhibits the stimuli of the vagal nerve, and, therefore, its anti-inflammatory properties result in negative effects on the GI tract [250]. Multiple studies showed that stress, anxiety, and depression are trigger factors for relapsing in IBD, and additionally, IBD patients are at higher risk of developing depression than healthy individuals [8,172,186,193,251].

Recent research suggests that psychological interventions can improve the treatment effect of GI diseases, thus improving patients’ quality of life [252,253,254,255]. Cognitive behavioral therapy (CBT) is considered the most effective psychotherapy for an IBD multidisciplinary approach among various psychological interventions. It can reduce the rate of psychological disorders and improve the quality of life of IBD patients [256,257,258,259]. Furthermore, Jordan et al. emphasized in their study that CBT significantly reduced anxiety and low mood scores and increased quality of life scores. It also reduced disease activity in the specific group of IBD patients who also experienced anxiety and low mood [257]. Similarly, treatment with CBT was associated with significantly greater improvement in depressive severity in the overall sample of young people with depression and CD and correlated with a significantly greater improvement in pediatric CD activity in the subgroup with active IBD [260]. However, a recent systematic review has shown that the positive effects of CBT regarding improving the mental state of IBD patients are not long-term. There are insufficient data to determine if CBT improves disease activity and reduces inflammation [261].

Another type of psychological intervention to reduce stress is mindfulness activities. This intervention is shown to have a positive role in stress level reduction and improving the quality of life in patients with IBD [262,263,264]. Additionally, mindfulness-based therapy reduces inflammation related to IBD by decreasing the levels of inflammatory biomarkers, such as interleukin-6, fecal calprotectin, and C reactive protein [264,265,266].

The multidisciplinary approach to IBD includes various psychotherapy interventions, such as gut-directed hypnotherapy, breath–body–mind workshops, advanced combination treatment, and psychological counseling. Gut-directed hypnotherapy is associated with improved GI function and may work through immune-mediated pathways in chronic diseases. It is demonstrated to have a significant effect of psychosocial therapy on extending clinical remission in patients with UC [267]. A recent randomized controlled trial showed that hypnotherapy was not superior to standard medical therapy in patients with IBD in remission with IBS-type symptoms [268]. Gut-directed hypnotherapy is an increasingly used therapy for patients with IBS and IBD. However, it remains to be investigated whether it can be used for first-choice adjuvant therapy [269]. Combining educational and psychological counseling with CBT improves a patient’s quality of life in both emotional and social functions [270]. Gerbarg et al. demonstrate that breath–body–mind workshops for patients with IBD are linked to significant enhancement in psychological and physical symptoms and higher quality of life scores [271]. Mind–body interventions can include participating in yoga classes related to reducing stress and increasing the ability to manage symptoms [272,273,274].

Potential adjunctive therapy involves combining psychotherapy with antidepressant medication. It was demonstrated that using serotonin–noradrenaline reuptake inhibitors, selective serotonin reuptake inhibitors, and tricyclic antidepressants has a positive influence by alleviating psychological and physical symptoms in patients with IBD and disease activity. Additionally, antidepressants improve sleep quality and chronic pain [275,276,277]. Despite the potential adverse effects of the medication, evidence suggests that antidepressant medication has a significant role in improving patients’ quality of life and mental health, notwithstanding the management of the disease [201,278].

## 14. Future Perspectives Regarding the Interaction between Stress and Inflammation

### 14.1. miR-129-5p—A Significant Controller of Different Pathways

In the past few years, research teams have extensively investigated the role of RNA molecules, particularly microRNAs (miRNAs), in the oncogenic processes related to digestion [279,280]. Cellular apoptosis exhibits tissue specificity, while miRNAs emerge as genuine biomarkers of the tumorigenesis phenomenon [279,281]. The importance of miRNAs in pathology is notable, given their responsiveness to stress and altered expression patterns as diseases progress. Extensive research has focused on understanding the roles and modifications of miRNAs in cancer development and progression, with potential implications for significant clinical and therapeutic advancements in molecular research. Additionally, in vitro studies highlight miRNAs as promising candidates for molecular replacement therapy, offering potential avenues to hinder cancer progression, prevent lymph node metastasis, and induce tumor cell apoptosis [282,283]. Furthermore, every investigation into miRNAs advances our understanding of the molecular mechanisms underlying oncogenesis and neurodegenerative disorders. Recent research underscores the significance of exploring miRNAs and their target genes for their prospective contributions to personalized treatment strategies across diverse diseases [279,282,283].

Given that circulating miRNAs serve as significant regulators in intercellular communication and exhibit stable expression patterns in both tissues and biological fluids, their clinical significance deserves highlighting. MiRNAs, non-coding molecules comprising 21 to 23 nucleotides, have garnered attention due to their presence in body fluids. Recent research underscores their potential as disease biomarkers for screening, monitoring disease progression, and predicting therapeutic outcomes [279]. Recent research [279,284] has underscored the significance of miRNAs in the pathogenesis of numerous diseases. It has been observed that a single miRNA can regulate multiple genes, while conversely, a single gene can be targeted by multiple miRNAs [284].

Post-transcriptional mechanisms allow for a specific miRNA to regulate up to roughly 60% of the protein-coding genes. Moreover, miRNAs are involved in vital cellular functions such as cell growth, homeostasis, apoptosis, and cell migration. In the realm of brain development, a solitary miRNA may participate in multiple processes, including synaptic formation and neural development. Contemporary molecular research is centered on identifying clinical alterations resulting from miRNA dysregulation to tailor targeted treatments for various diseases [279,280,282,284]. Research indicates that miRNAs play a pivotal role in neurological disorders by modulating both the inflammatory response and nerve injury [279,285]. This offers fresh perspectives for understanding degenerative diseases and neural damage. For instance, recent studies have focused on the detrimental aspects of Alzheimer’s disease (AD) [279,286], the primary cause of dementia. AD is characterized by the accumulation of beta-amyloid aggregates, leading to neuroinflammation and brain damage. Currently, the molecular alterations in AD are not fully understood, and recent research has aimed to elucidate the role of miRNAs and their implications in tau hyperphosphorylation, which contributes to the formation of neurofibrillary tangles and amyloid plaques [286].

Given the escalating incidence of neurodegenerative diseases, there is a demand for novel biomarkers to facilitate early diagnosis and innovative treatment avenues. MiRNAs present an opportunity to uncover molecular alterations in neurodegenerative disorders, underscoring the necessity for additional research to identify promising biomarkers for early detection. Furthermore, exploring the involved pathways and targeted genes is imperative in advancing this field.

A recent review article [279] underscored the pivotal role of miR-129-5p as a central regulator across diverse disease pathways. Notably, significant pathways implicated include WNT and PI3K/AKT/mTOR, whose aberrations contribute to digestive neoplasia and neurodegenerative disorders. Furthermore, miR-129-5p has been implicated in modulating transmembrane metalloproteinases, integrins, and high-mobility group protein B1 (HMGB1), with its heightened expression providing a shield against cell proliferation and metastasis in digestive cancer cases. The review also elucidates how acute and chronic stress contribute to the dysregulation of this miRNA, culminating in neuroinflammation, neural apoptosis, and the emergence of depression [279]. Given that both acute and chronic stress can potentially lead to the dysregulation of miR-129-5p, the authors observed a correlation between molecular regulation in the brain and the digestive tract [279]. The authors inferred that stress-induced downregulation of this miRNA could lead to the suppression of HMGB1 inhibition, consequently promoting tumor proliferation [279]. The heightened expression of miR-129-5p disrupts pathways involving various targeted genes, offering protective and therapeutic potential against numerous diseases. Moreover, while miR-129-5p has been extensively researched and assessed in various gastrointestinal cancers, it also serves as a significant indicator in brain development, neurodegenerative diseases, depression, and responses to both chronic and acute stress [287].

### 14.2. Fecal Microbiota Transplantation in Inflammatory Bowel Disease

In recent years, fecal microbiota transplantation (FMT), a treatment centered on the microbiome, has garnered significant attention across scientific, clinical, and public audiences [288]. Using a range of techniques and methods, FMT is a complex procedure aimed at reinstating a harmonious intestinal flora. It involves infusing feces from healthy donors into the GI of individuals with certain conditions to promote recovery. FMT has demonstrated effectiveness in treating various non-GI and GI diseases, including idiopathic constipation, recurrent Clostridioides difficile infection, IBD, and IBS [289,290].

FMT is not a novel therapeutic concept; however, it has garnered heightened interest in recent years due to advancements in methodology and expanding clinical applications [291].

The severity of IBD and its complications are positively correlated with an overgrowth of pathogenic bacteria, including Coprobacillus, Clostridium ramosum, and Clostridium hathewayi. Conversely, a decrease in beneficial anti-inflammatory bacteria like Faecalibacterium prausnitzii is observed, which typically regulates the host’s immune system within the gut. These protective bacteria play a crucial role in immunosuppression, thereby preventing the induction of cytokines and potential intestinal damage [292]. Among patients with CD, their microbiota exhibited a predominance of Actinomyces spp., along with elevated levels of *Veillonella* spp., *E. coli*, and *Intestinibacter* spp. Conversely, in patients with UC, the gut microbiota showed an augmentation of Eubacterium rectum, *E. coli*, and *Ruminococcus gnavus*, which are microbes known to sustain and trigger cellular inflammation [292,293].

FMT is emerging as a groundbreaking approach for managing severe cases of IBD, demonstrating a significant success rate. This procedure involves delivering a healthy fecal solution into the recipient’s intestinal tract. Studies have shown that autologous FMT (a-FMT) yields comparable benefits to heterologous FMT (h-FMT) [289,292].

The objective of a-FMT is to restore the disrupted gut microbial community by utilizing one’s feces in a healthy condition. Meanwhile, h-FMT involves transplanting feces from a healthy donor into the affected individual to address conditions like IBD and other infectious diseases. It is generally preferred to opt for a-FMT over h-FMT to mitigate the risk of infectious complications [294,295]; nonetheless, it is crucial to identify stool samples that are functionally optimal to mitigate complications associated with inflammation in IBD. Enhancing the efficacy of FMT in IBD necessitates consideration of various factors, including donor selection criteria, the recipient’s current disease status, and the standardization of processing protocols [296].

The ideal timing for conducting the transplant continues to be a topic of debate, with doctors lacking consensus on this matter. While greater effectiveness of transplantation has been noted in severe cases of IBD, the procedure is also advocated for individuals newly diagnosed with the condition. FMT seems to offer both safety and success in averting recurrent infections among individuals with IBD [297]. Due to the inclusion of various factors and the absence of established protocols, conducting studies on FMT can be challenging. Despite its promise and the new avenues it has opened in research, it will likely be a considerable amount of time before this treatment method becomes standard practice in hospitals worldwide [298].

## 15. Discussion

CD and UC are long-lasting debilitating conditions linked to psychological and social complications. Adolescents are particularly susceptible to the additional stress of dealing with IBD while navigating important developmental stages. Psychological and social factors, including catastrophizing, the stigma surrounding the illness, uncertainty about the condition, and feelings of shame associated with the illness, frequently contribute to perceived stress in chronic illnesses. Nonetheless, the impact of these variables on perceived stress among adolescents with IBD remains unexplored [244].

Recent clinical research studies have also shown that stress is linked to gastrointestinal health and problems with digestion [133]. A phylogenetic microarray, for instance, was used in one study to demonstrate that stress exposure during pregnancy led to abnormal microbiota colonization patterns in children, which probably enhanced inflammation and gastrointestinal complaints [118]. According to these findings, enhanced bacterial translocation was also linked to stress-related psychiatric diseases including depression, which in turn activated immune responses against commensal bacteria [134]. While the data suggest a significant impact of stress on the intestinal microbiota, it is important to note that stress is a subjective phenomenon. This subjectivity poses a challenge when attempting to objectively assess the effects of stress. Consequently, additional studies involving human participants are necessary to confirm whether stress indeed leads to dysbiosis of the gut microbiota.

IBD was categorized as a psychosomatic condition in the 1950s after numerous early research studies revealed a connection between IBD and mental diagnoses [123]. In numerous IBD treatment trials [24,25,26], placebo response rates might still reach 30–40%, so these results support the idea that changes in one’s psychological state can alter disease activity. Acute psychological stress has an impact on water and ion secretion as well as gastrointestinal motility. Acute short-term stress, such as stressful interviews, dichotomous listening exams, and painful stimuli, increases colonic motility in healthy human volunteers and enhances the secretion of salt, chloride, and jejunal water. Although these are non-inflammatory changes, stress-related elevations in IBD patients’ symptomatology may be a result of them [299,300,301].

Acute psychological stress also has an impact on mucosal inflammation. It has been demonstrated that the central release of neuropeptide SP from afferent neurons plays a critical role in modulating stress-induced gastrointestinal hyperalgesia. Peripheral release of SP from the ENS may contribute to stress-related increases in mucosal inflammation in addition to its central effects. Mast cells, a cell type regarded as crucial in mediating stress-induced permeability changes, are found in close connection with SP-containing neurons, even though there are no published data that show an increase in mucosal SP in response to stress. It has been demonstrated that SP causes IBD patients’ mucosal mast cells to release more histamine. Finally, SP can operate as a neurotransmitter as well as an independent inflammatory cytokine, increasing cytokine production and promoting the migration of inflammatory cells. Additionally, it stimulates the expression of CD11b on neutrophils and leucocyte adhesion molecules on microvascular endothelium, promoting leucocyte adherence at inflammatory locations [153,302].

Stress has been linked to increased disease activity [303] and decreased quality of life [304] in individuals with IBD. Since the 1930s [305], it has been suspected to contribute to the onset of the disease and serve as a potential trigger for disease flares [240,306]. Previous reviews have found a significant association between stress and IBD disease activity in the majority of studies reviewed. However, concerns have been raised regarding the heterogeneity in study design, participant samples, and measures of disease activity and stress, which have hindered the establishment of a clear relationship between stress and disease activity in IBD [174,307,308].

Given the recent promotion of collaborative care models for IBD [309], it is crucial to have a comprehensive understanding of the burden of psychiatric comorbidity in IBD to allocate appropriate resources. Further research is necessary to uncover the underlying causes of these associations and to optimize the treatment of mental disorders in individuals with IBD. Clinicians should remain highly vigilant in identifying and treating these psychiatric associations, as it can benefit individuals with IBD both in terms of their mental health and potentially impact the course of their disease.

IBD often occurs at younger ages, thus interfering with one’s education, career, or daily routine. IBD negatively influences quality of life, with its restrictions on activities, interpersonal relationships, and well-being. The burden of IBD begins with symptoms and extends to mental well-being. IBD uniquely impacts an individual’s life, but the influence crosses borders and affects the entire family and society [204].

## 16. Conclusions

A lot of contributing factors can influence the course of the disease over time for a patient diagnosed with IBD. Psychological stress and psychosocial impairment conditions can exacerbate the intestinal inflammatory response, increasing the likelihood of being diagnosed with IBD in the following years. Current research has shown that exposure to high levels of stress can increase relapse rates in patients with quiescent forms of IBD. Reducing exposure to stress can reduce the severity of symptoms in patients diagnosed with IBD, improving their quality of life. The early and correct approach to the course of somatic and psychological conditions can improve the analysis of their interdependence on the disease state.

It is well known that stress is a factor that accelerates the progression of IBD. Therefore, psychological interventions can improve the treatment effect of gastrointestinal diseases, thus improving patients’ quality of life. There are multiple interdisciplinary approaches to IBD such as cognitive behavioral therapy, mindfulness-based therapy, breath–body–mind workshops, or gut-directed hypnotherapy. Management of these chronic diseases (CD and UC) should be designed individually, respecting medical interventions. Areas of adjuvant therapies have arisen to treat comorbidities and to increase patient quality of life. Thus, the need for future research into preventing IBD and healthcare innovations to manage these complex and costly diseases is highlighted.

## 17. The Limitations and Strengths of this Work

A limitation of this work is related to the fact that this review relies heavily on animal studies and preclinical research to support its arguments. While animal studies can provide valuable insights, their relevance to human physiology and pathology may be limited. However, this is one of the few published reviews describing the interplay between stress and inflammatory bowel diseases not only in adult patients but also in children. This study also addresses the role of early-life stress and its potential impact on neonatal gastrointestinal health, with an emphasis on NEC and its association with stress in preterm infants.

## Figures and Tables

**Figure 1 jcm-13-01361-f001:**
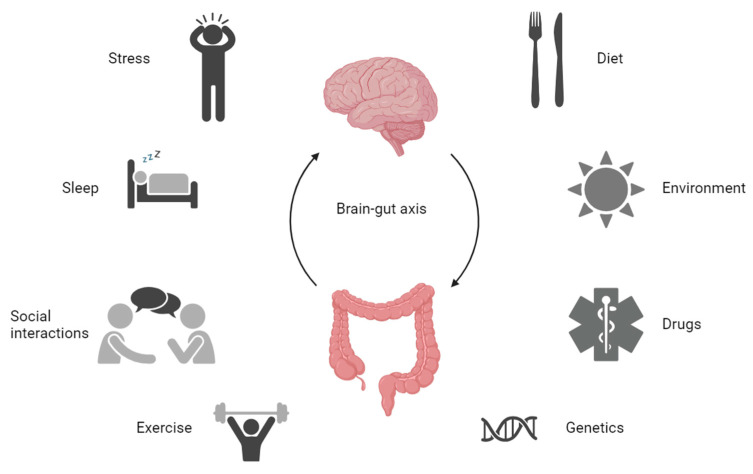
Schematic representation of the brain–gut axis.

**Figure 2 jcm-13-01361-f002:**
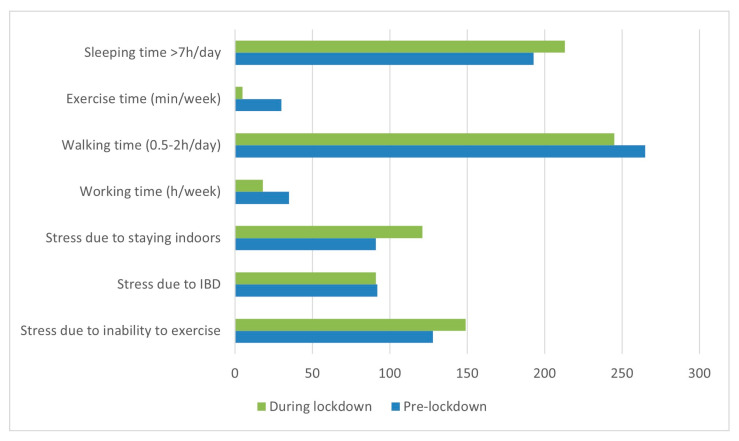
Comparison of psychological stress and lifestyle factors before and during lockdown in IBD patients.

**Figure 3 jcm-13-01361-f003:**
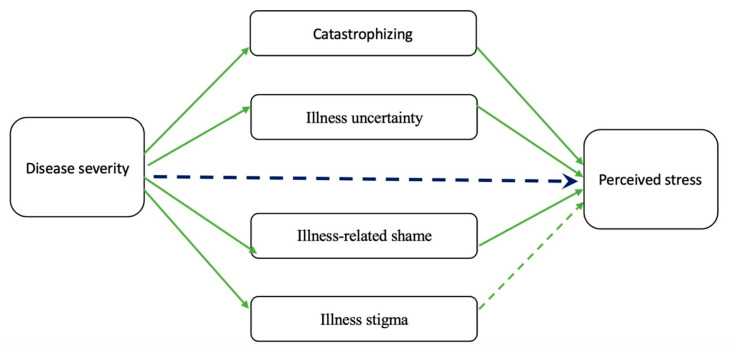
Psychosocial variables (catastrophizing, illness stigma, illness uncertainty, and illness shame). Legend: solid lines represent significant effects, dotted lines represent nonsignificant effects; green represents indirect effects, and blue represents direct effects.

**Table 1 jcm-13-01361-t001:** Research exploring the correlation between stress and the composition of the gut microbiota.

Factor	Author(s)	Type of Study	*N*	Intervention/Methodology	Result
Pro-/prebiotics	Messaoudi, M. et al. [107]	In vivo rat study	36 rats	Probiotic formulation	Observed anxiolytic-like effects in rat models
	Garcia-Rodenas, CL. et al. [108]	In vivo rat study	84	Maternal separation and prebiotics/probiotics/LC-PUFA	Implementation of a nutritional intervention during the weaning period resulted in the normalization of gut permeability and the restoration of the growth rate
	Zareie, M. et al. [109]	In vivo rat study	Four to five rats per group	WAS and probiotics	Probiotics exhibited a preventive effect on chronic stress-induced gastrointestinal abnormalities
	Li, N. et al. [110]	In vivo mouse study	Eight mice per group	Chronic mild stress and probiotics	Reduced levels of pro-inflammatory cytokines and modified stress-induced behavioral patterns
	Bravo, J.A. et al. [111]	In vivo mouse study	36	Probiotic formulation	Highlighted the significance of probiotics in the bidirectional communication between the gut and the brain in stress-related disorders
	Messaoudi, M. et al. [107]	Double-blind, placebo-controlled, randomized parallel group study	66 individuals	Probiotic formulation	Evidenced favorable psychological outcomes in a cohort of healthy human volunteers
	Rao, S. et al. [112]	Systematic review	11 RCTs	Prebiotic supplementation	Demonstrated transient advantageous effects on the composition of the intestinal microbiota in the short term
Prenatal/early life stress	O’Mahony, S.M. et al. [113]	In vivo rat study	22	Maternal separation	The effects of early life stress on the gut–brain axis led to modifications that contributed to the manifestation of symptoms in IBD
	Golubeva, A.V. et al. [114]	In vivo rat study	6–10 per group	Prenatal stress	Persistent modifications in the composition of the intestinal microbiota over an extended period
	Jasarevic, E. et al. [115]	In vivo mouse study	21–23 mice per group	Prenatal stress	Changes in the vaginal microbiota were implicated in the process of reprogramming the developing brain
	Xie, R. et al. [116]	In vivo mouse study	6–20 mice per group	Maternal high-fat diet	Intestinal dysbiosis and the presence of chronic low-grade inflammation in the gastrointestinal tract
	Bailey, M.T. et al. [117]	In vivo primate study	20	Maternal separation	Psychological disturbances resulting from maternal separation led to modifications in the composition of the intestinal microflora
	Zijlmans, M.A. et al. [118]	Longitudinal clinical study	192 children	Questionnaire	The presence of prenatal stress was correlated with specific microbial colonization patterns in infants
Chronic/social/environmental stress	Soderholm, J.D. et al. [119]	In vivo rat study	Seven to eight rats per group	WAS	Impaired mucosal defenses against luminal bacteria lead to intestinal inflammation
	Da Silva, S. et al. [120]	In vivo rat study	13–14 rats per group	WAS	Modified composition of the intestinal mucus
	Meddings, J.B. et al. [121]	In vivo rat study	Not specified	Stress induction	Elevated gastrointestinal permeability facilitates the passage of luminal constituents to the mucosal immune system
	Saunders, P.R. et al. [122]	In vivo rat study	6	Cold-restraint stress or WAS	Intensified intestinal inflammation resulting from enhanced uptake of immunogenic substances
	Santos, J. et al. [123]	In vivo rat study	Four rats per group	WAS	Epithelial mitochondrial damage triggered by stress and activation of mucosal mast cells
	Gao, X. et al. [124]	In vivo mouse study	Four to six mice per group	Chronic restraint stress	Disrupted gut microbiota followed by immune system activation resulted in the development of colitis
	Neufeld, K.M. et al. [125]	In vivo mouse study	12 mice per group	Germ-free and specific-pathogen-free	The presence of typical intestinal microbiota played a role in the development of behavior
	Donnet-Hughes, A. et al. [126]	In vivo mouse study	10 mice per group	Lactation	Cellular transfer of bacterial translocation took place in mice during pregnancy and lactation
	Heijtz, R. et al. [127]	In vivo mouse study	7–14 mice per group	Germ-free and specific-pathogen-free	The gut microbiota impacted the development of the mammalian brain and subsequent behavioral patterns in adulthood
	Marin, I.A. et al. [128]	In vivo mouse study	10–12 (three independent experiments)	Unpredictable chronic mild stress	Modified composition of the intestinal microbiota, particularly within the lactobacillus component
	Bharwani, A. et al. [129]	In vivo mouse study	7–20 mice per group	Chronic social defeat	Stress triggered intricate structural alterations in the gut microbiota
	Sudo, N. et al. [36]	In vivo mouse study	18–24 mice per group	Germ-free and specific-pathogen-free; acute restraint stress	The commensal microbiota had the potential to influence the postnatal maturation of the hypothalamic–pituitary–adrenal (HPA) stress response
	Galley, J.D. et al. [130]	In vivo mouse study	Five mice per group	SDR	Altered microbial populations that had a close association with the colonic mucosa
	Bailey, M.T. et al. [33]	In vivo mouse study	Five mice per group	SDR	Stress resulted in notable alterations in the colonization of the intestinal microbiota
	Noguera, J.C. et al. [131]	Field experiment in wild birds	64	Corticosterone implant	Modified gut microbiome in birds living in their natural habitat
	Van der Zaag-Loonen, H.J. et al. [132]	Clinical study	65	Coping style instrument	Adolescents with IBD exhibited a higher utilization of avoidant coping strategies in comparison with healthy individuals
	Walker, L.S. et al. [133]	Clinical study	263	Daily interview assessment	There was an association between stress and the occurrence of digestive problems and disruptions in gastrointestinal health
	Maes, M. et al. [134]	Clinical study	40	Depression	Elevated bacterial translocation and heightened immune responses targeting commensal bacteria

HPA = hypothalamic–pituitary–adrenal axis; LC-PUFAs = long-chain poly-unsaturated fatty acids; RCT = randomized controlled trial; SDR = social disruption; WAS = water-avoidance stress.

## Data Availability

The data presented in this study are available upon request from the corresponding author.

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
