# Peer review of "The Interaction between Stress and Inflammatory Bowel Disease in Pediatric and Adult Patients"

_jcm, 2024, doi:10.3390/jcm13051361_

Round 1

Reviewer 1 Report

Comments and Suggestions for Authors

The literature review, "The Interaction Between Stress and Inflammatory Bowel Disease in Pediatric and Adult Patients," is a valuable work that brings relevant information from the specialized literature to the forefront. Nevertheless, some modifications should be made:

1.       The materials and methods chapter should clearly present and elaborate on the research strategy.

2.       Chapter 3 should include a graphical presentation of the gut microbiota-brain axis.

3.       A chapter related to future perspectives in this research area should be added. Multiple recent articles address topics of interest in the presented theme, specifically – implications related to microRNAs (more details can be found in the article https://doi.org/10.3390/biomedicines11072058) and, of course, microbiota transfer as a potential treatment (more details and relevant articles can be found at https://doi.org/10.3390/biomedicines11041016 ).

4.       The discussion chapter is practically nonexistent. Given the substantial size of the work, this chapter is necessary to discuss contradictions in the literature and to highlight the most important aspects of the review that require detailed comparison with themes from the literature.

5.       The conclusions chapter needs to be expanded and modified to encompass most of the review's findings.

6.       The bibliography needs to be adapted according to MDPI standards.

Comments on the Quality of English Language

Minor editing of English language required

Author Response

Dear Reviewer,

Thank you for your feedback. We would like to respond to your comments:

  1. The authors have clearly presented the research strategy in the materials and methods chapter.
  2. We included a graphical presentation of the gut microbiota-brain axis.
  3. The authors introduced a chapter related to future perspectives in this research area (chapter 14).
  4. The authors introduced the discussion chapter.
  5. The authors expanded the conclusions chapter.
  6. The authors adapted the bibliography according to MDPI standards.

Kind regards,

Dr. Basaca

Reviewer 2 Report

Comments and Suggestions for Authors

Nice review. I have comments to improve it:
1. Your search date is up to December 2022. Please update it to 2023 and reassess the new articles identified.
2. Page 5: Cite a relevant review regarding the association between IBD and metabolic conditions (https://pubmed.ncbi.nlm.nih.gov/35274325/).
3. You should open a new paragraph and cite and discuss the Mendelian randomization studies evaluating the causal associations between IBD and psychologic conditions (https://www.ncbi.nlm.nih.gov/pmc/articles/PMC10725559/). This would be a strong point for your review article.

Moreover,

  1. The study needs clear organization and coherence. It jumps between various topics related to stress, gut microbiota, and inflammatory bowel disease without a smooth transition or clear delineation of each section's purpose. This makes it challenging for readers to follow the argument effectively.
  2. While the study discusses the potential mechanisms underlying the interaction between stress and IBD, it fails to provide concrete evidence or references to support its claims. Statements like "More and more proof indicates..." or "Some studies link..." without specific citations diminish the study's credibility and leave the reader questioning the validity of the assertions.
  3. The language used in the study is overly technical and lacks clarity. Concepts such as the hypothalamic-pituitary-adrenal (HPA) axis and cholinergic anti-inflammatory system are introduced without sufficient explanation for readers who may not be familiar with these terms.
  4. The study relies heavily on animal studies and preclinical research to support its arguments. While animal studies can provide valuable insights, their relevance to human physiology and pathology may be limited.
  5. The discussion of early life stress and its potential impact on neonatal gastrointestinal health is intriguing but needs more depth. The study briefly mentions NEC and its association with stress in preterm infants. However, it needs to comprehensively explore the mechanisms underlying this association or discuss potential interventions to mitigate the effects of early life stress on gastrointestinal health.
  6. The study's conclusion and implications for clinical practice could be more specific and developed. While the study highlights the potential link between stress, gut microbiota, and inflammatory bowel disease, it does not offer practical recommendations for clinicians or insights into how this knowledge could inform the development of novel therapeutic approaches or interventions for patients with inflammatory bowel disease.
  7. What were your limitations and strengths?

Author Response

Dear Reviewer,

Thank you for your feedback. In response to your comments:

  1. The authors updated the research up to December 2023 and reassessed the new articles identified.
  2. The authors cited at line 208 the review regarding the association between IBD and metabolic conditions.

  3. The authors added a paragraph and discussed the Mendelian randomization studies evaluating the causal associations between IBD and psychologic conditions.

  4. The authors made a smooth transition between different sub-chapters and explained that there is a bidirectional interplay between stress and gut microbiota in patients with inflammatory bowel disease.
  5. The authors added in the manuscript specific citations to each statement like "More and more proof indicates..." or "Some studies link..." that were missing references, to support its claims.
  6. The authors explained in the text the concepts like hypothalamic-pituitary-adrenal (HPA) axis or cholinergic anti-inflammatory system for readers who may not be familiar with these terms.
  7. The authors added at the end of the manuscript a paragraph regarding the limitations of the work.

    A limitation of this work is related to the fact that this review relies heavily on animal studies and preclinical research to support its arguments. While animal studies can provide valuable insights, their relevance to human physiology and pathology may be limited.
  8. The authors added new data regarding early life stress and its potential impact on neonatal gastrointestinal health.
  9. The authors updated the conclusions introducing recommendations related to potential interventions to mitigate the effects of stress on gastrointestinal tract in patients with IBD.
  10. The authors added at the end of the manuscript a paragraph regarding the limitations and strengths of the work.

Kind regards,

Dr. Basaca

Round 2

Reviewer 1 Report

Comments and Suggestions for Authors

The authors have implemented all requested changes.

Comments on the Quality of English Language

Minor editing of English language required.

Reviewer 2 Report

Comments and Suggestions for Authors

Thanks for your revisions.